# Novel Systemic Treatment Modalities Including Immunotherapy and Molecular Targeted Therapy for Recurrent and Metastatic Head and Neck Squamous Cell Carcinoma

**DOI:** 10.3390/ijms23147889

**Published:** 2022-07-17

**Authors:** Soma Ghosh, Pooja A. Shah, Faye M. Johnson

**Affiliations:** 1Department of Thoracic/Head & Neck Medical Oncology, The University of Texas MD Anderson Cancer Center, 1515 Holcombe Blvd., Box 432, Houston, TX 77030, USA; sghosh5@mdanderson.org (S.G.); pshah3@mdanderson.org (P.A.S.); 2Graduate School of Biomedical Sciences, The University of Texas, Houston, TX 77030, USA

**Keywords:** head and neck squamous cell carcinoma, human papillomavirus, novel targeted therapy, cetuximab, immune-checkpoint inhibitor

## Abstract

Head and neck squamous cell carcinomas (HNSCCs) are the sixth most common cancers worldwide. More than half of patients with HNSCC eventually experience disease recurrence and/or metastasis, which can threaten their long-term survival. HNSCCs located in the oral cavity and larynx are usually associated with tobacco and/or alcohol use, whereas human papillomavirus (HPV) infection, particularly HPV16 infection, is increasingly recognized as a cause of oropharyngeal HNSCC. Despite clinical, histologic, and molecular differences between HPV-positive and HPV-negative HNSCCs, current treatment approaches are the same. For recurrent disease, these strategies include chemotherapy, immunotherapy with PD-1-inhibitors, or a monoclonal antibody, cetuximab, that targets epidermal growth factor; these therapies can be administered either as single agents or in combination. However, these treatment strategies carry a high risk of toxic side effects; therefore, more effective and less toxic treatments are needed. The landscape of HNSCC therapy is changing significantly; numerous clinical trials are underway to test novel therapeutic options like adaptive cellular therapy, antibody-drug conjugates, new targeted therapy agents, novel immunotherapy combinations, and therapeutic vaccines. This review helps in understanding the various developments in HNSCC therapy and sheds light on the path ahead in terms of further research in this field.

## 1. Introduction

Head and neck squamous cell carcinomas (HNSCCs) originate in the oral cavity, pharynx, larynx, and sinonasal track. Squamous cell carcinoma is the most common type of cancer in the head and neck region and the sixth leading cancer by incidence worldwide [1]. HNSCC accounts for about 5% of all cancers in the United States [2]. Many risk factors for HNSCC have been identified including exposure to carcinogens (such as tobacco, alcohol, betel nut, and air pollution [3]), infection with the oncoviruses Epstein–Barr virus or high-risk human papillomavirus (HPV) strains [4,5], and genetic factors. Several of these risk factors display geographical variations; for instance, betel nut chewing is most common in India, while exposure to carcinogenic air pollutants is more common in developing regions, such as India and China [6,7]. The incidence of HPV-driven HNSCC is increasing in Western countries, whereas Epstein–Barr virus-driven HNSCC is more common in developing countries in East Asia. One of the main genetic disorders that raises the risk of developing HNSCC is Fanconi anemia. This rare inherited disease is caused by mutations in one or more of the *FANC* family of DNA-repair genes. Individuals with Fanconi anemia have a 500- to 700-fold higher risk of developing HNSCC than the general population [8]. Other distinct factors that separate individuals with FA from the general population in terms of HNSCC include diagnosis at a young age (20–40 years old) and tumor localization in the oral cavity and tongue.

HPV-driven HNSCC occurs predominantly in the oropharynx; the primary oncogenic HPV strains are HPV16 (83–86% of HPV-positive HNSCCs), HPV33 (3.3–7.3%), HPV35 (2.2–4%), and HPV18 (<2%) [9,10,11,12]. In contrast, carcinogen exposure typically drives HPV-negative disease. HPV-positive HNSCC occurs in a younger patient population and has a more favorable prognosis than HPV-negative HNSCC. For patients with advanced-stage HPV-positive HNSCC, the 5-year survival rates are 75% to 80%, whereas fewer than 50% of patients with HPV-negative disease survive for 5 years [13]. Although HPV-positive and HPV-negative HNSCC distinctly differ genetically, they are treated in much the same way, an approach that produces significant morbidity [14].

In terms of the screening strategy for early-stage HNSCC, visual screening has been considered as a feasible, safe, and cost-effective option in the last few years amongst the high-risk group of tobacco, betel, and/or alcohol consumers [15]. 

In general, early-stage HNSCC is treated with surgery or radiation and has 5-year survival rates of approximately 70% to 90% [16]. Locally advanced disease requires multimodal treatment that combines surgery, radiation, and systemic treatment with platinum-based chemotherapy or anti-epidermal growth factor receptor (EGFR) targeted therapy with cetuximab [17,18,19]. 

Local recurrence or the development of distant metastasis is common in HNSCC, affecting about 20% of patients treated for early-stage disease and 50% of those with locally advanced HNSCC [16]. The prognosis is poor for recurrent or metastatic (R/M) HNSCC, with a median duration of around 1 year of overall survival (OS) [17]. R/M HNSCC that is not treatable with surgical resection or definitive radiotherapy is treated with palliative systemic therapy that includes platinum-based chemotherapy, cetuximab, and/or immune checkpoint inhibitors (ICIs) with anti-programmed death 1 (PD-1) antibodies (Figure 1). Conventional therapy for locally advanced HNSCC often results in permanent impairments in chewing, swallowing, and tasting, along with a dry mouth, feeding tube dependence, and aspiration pneumonia [20,21]. These adverse events for survivors, coupled with the poor outcomes for R/M HNSCC, demonstrate the need for novel therapies with less toxicity and more efficacy. Several novel therapies—including molecular targeted therapies, antibody-drug conjugates, and immunotherapies—may be more selective, cause fewer adverse effects, and be more effective in the treatment of HNSCC. In our review, we describe recent developments in the understanding of the genomics and pathophysiology of HNSCC, assess progress in the management of HNSCC, and provide perspectives on future research and treatment directions.

## 2. Genomics of HNSCC

Genomic studies in HNSCC have revealed frequent chromosomal changes, DNA copy number alterations, somatic mutations, and promoter methylation. Only a few of the mutations and chromosomal abnormalities driving HNSCC were known before the introduction of next-generation sequencing (NGS) [22,23]: *TP53* and *CDK2NA* [24] mutations and amplification of 11q13, *CCND1*, and *EGFR* [25]. The first results of NGS studies in HNSCC [26,27] and The Cancer Genome Atlas (TCGA) [12] have offered a comprehensive understanding of the somatic genomic alterations driving HNSCC. In addition to previously known HNSCC-associated mutations in *TP53*, *CDKN2A*, and *PIK3CA*, these studies identified *NOTCH1* as one of the most commonly mutated genes in HNSCC [28]. The topmost frequently mutated genes in HNSCC are *TP53* (72%), *CDKN2A* (22%), *FAT1* (23%), *PIK3CA* (21%), *NOTCH1* (19%), *KMT2D (MLL2)* (18%), *NSD1* (10%), *CASP8* (9%), *AJUBA* (6%), and *NFE2L2* (6%). HNSCCs infrequently carry RAS gene mutations; mutations in *HRAS*, at about 8%, are the most common in this group. Strikingly, NGS revealed that unlike many other solid tumors driven by mutations in oncogenes, HNSCCs are most often characterized by the loss of tumor-suppressor genes. 

HPV-positive HNSCCs, however, typically lack mutations or alterations in *TP53* and *CDKN2A* genes. Instead, these tumors more commonly have a *PIK3CA* genomic alteration with mutation (14%) and gene amplification (16%) [12], loss of *TRAF3,* and amplification of *E2F1* [27]. NGS of 149 HPV-positive and 335 HPV-negative HNSCC/normal paired samples by Gillison et al. [29] confirmed *PIK3CA* mutations and identified mutations in *ZNF750* and *EP300* as candidate driver events in HPV-positive HNSCC. The outcome of the study was in line with earlier findings that showed *TP53*, *CDKN2A*, *FAT1*, *CASP8*, *NOTCH1*, and *HRAS* as the main mutations that drove cancer progression in HPV-negative HNSCC. Furthermore, a recent study also showed that high-risk HPV infections went together with mutations in *PIK3CA*, *EP300*, *NF1*, and *RB1* in samples from benign tonsils, suggesting these mutations to be potential biomarkers to identify the cancer progression risk [30].

Substantial research has also confirmed the role of aberrant signaling pathway activation in HNSCC. For instance, EGFR is frequently expressed in 80% to 90% of HNSCC tumors, and its overexpression is associated with poor survival [31]. Other receptor tyrosine kinases, such as HER2 and MET, are also overexpressed in many HNSCCs and may contribute to resistance to EGFR-targeting drugs [14,32]. In addition, the PI3K-AKT-mTOR signaling pathway, which drives the development of many tumor types, is often altered in HNSCC [12]. 

## 3. Pathophysiology of HNSCC

### 3.1. HPV-Negative HNSCC

HPV-negative HNSCC is typically observed in patients with regular consumption of tobacco and alcohol [33,34]. The carcinogenic effects of tobacco and alcohol are attributable to the formation of DNA adducts, which dysregulate critical cellular processes [35]. DNA adducts can cause chromosomal mutation and instability, affecting cell homeostasis and other cellular mechanisms, which leads to cancer progression [35]. The carcinogenic effects of tobacco can also be enhanced by alcohol, which can facilitate the introduction of tobacco-related carcinogens in the mucosa [36,37]. In this way, the collaborative effect of both alcohol and tobacco contributes to the HNSCC carcinogenesis [38]. 

### 3.2. HPV-Positive HNSCC

HPV is a nonenveloped, circular, double-stranded DNA virus that causes approximately 70% of oropharyngeal squamous cell carcinoma in the United States [39]. HPV strains can be classified by their associated cancer risk into low- and high-risk groups. The high-risk strains include HPV16, HPV18, HPV26, HPV33, HPV35, and HPV59, along with seven others. HPV16 is the primary strain causing HNSCC in the oropharynx (83%) [11]. The E6 and E7 HPV oncoproteins degrade the tumor suppressor TP53 and retinoblastoma-associated protein (RB) on infection [40]. In turn, the inactivation of Rb function by E7 leads to an increase in p16INK4A (p16) levels. P16 expression is commonly used as a surrogate marker for HPV-related oropharyngeal tumors [24]. Tumors can only be sustained with persistent expression of the viral E6 and E7 oncoproteins.

## 4. Current Targeted Therapy for Head and Neck Cancer

### 4.1. EGFR Inhibitors

EGFR is overexpressed in ~80% of HNSCCs, leading to a poor prognosis [41,42]. Monoclonal antibodies (mAb), as well as TKI-based small molecules, can be used to target EGFR. Cetuximab, a chimeric mAb, is an FDA-approved targeted therapy for HNSCC that blocks EGFR signaling [14,43].

An early phase-III randomized trial in locally advanced HNSCC that compared radiation plus cetuximab to radiation alone showed a significantly better median duration of response (24.4 vs. 14.9 months, *p* = 0.005), progression-free survival (PFS) (17.1 vs. 12.4 months, *p* = 0.006), and OS (49 vs. 29.3 months, *p* = 0.006) in patients receiving cetuximab compared to those receiving radiation alone [44]. Yet, the addition of cetuximab led to increased rates of rash and infusion reactions compared to radiotherapy alone. Furthermore, two phase-III clinical trials found that, in patients with HPV-positive HNSCC, cetuximab was inferior to the combination of cisplatin with radiotherapy [45,46].

Single-agent cetuximab had low efficacy in R/M HNSCC, with overall response rates (ORRs) ranging from 10% to 13% [47]. The reason for the poor efficacy of cetuximab may be attributed to the ErbB protein and ligand aberrations and/or activation of other downstream signaling components [48]. Vermorken et al. [49] evaluated the effect of cetuximab alone on 103 HNSCC platinum-refractory patients. None of the patients showed a complete response (CR). The disease control rate (DCR) was 46% and only 13% of patients showed a partial response (PR).

In the EXTREME trial [49], R/M HNSCC patients were randomized to doublet chemotherapy (cisplatin or carboplatin plus 5-fluorouracil) vs. the same chemotherapy plus cetuximab for their first-line therapy. The median OS was 7.4 vs. 10.1 months, respectively (hazard ratio [HR] 0.80; *p* = 0.04). The median PFS was 3.3 vs. 5.6 months, respectively (HR 0.54; *p* < 0.001). The addition of cetuximab increased the response rate from 20% to 36% (*p* < 0.001). As expected, toxicity was worse in the chemotherapy plus cetuximab group, with more sepsis, anorexia, and skin and infusion-related reactions.

Panitumumab is a fully human immunoglobulin G2 (IgG2) EGFR mAb that has lower immunogenicity than cetuximab. The phase-III SPECTRUM trial of panitumumab combined with cisplatin and 5-fluorouracil as a front-line treatment for R/M HNSCC showed that the median PFS was longer in the group that received panitumumab (5.8 vs. 4.6 months, *p* = 0.0036). However, there was no significant difference in the median OS (11.1 vs. 9 months, *p* = 0.14). Several grade 3 or 4 adverse events were more common in the panitumumab group [50]. Zalutumumab is a high-affinity human IgG1 mAb to EGFR. A phase-III trial found no significant difference in the median OS between patients with R/M HNSCC who received zalutumumab plus the best supportive care and those who received just the best supportive care (6.7 vs. 5.2 months, *p* = 0.0648). Neither panitumumab nor zalutumumab is approved for the treatment of R/M HNSCC.

EGFR TKIs have also been studied in HNSCC. A phase-III study (LUX-Head & Neck 1) compared gefitinib to methotrexate for R/M HNSCC and found similar median OS durations (5.6 vs. 6.7 months) [51]. Given the lack of convincing evidence to support its use, gefitinib has not been further developed for HNSCC. In patients with platinum-resistant R/M HNSCC, afatinib (EGFR TKI) and cetuximab had similar response rates [52]. A phase-III trial comparing afatinib to methotrexate in patients with R/M HNSCC who had disease progression after platinum-based chemotherapy found median OS durations of 6.8 months with afatinib and 6 months with methotrexate (HR 0.96, *p* = 0.7) [53]. Although both LUX-Head & Neck 1 and 3 showed a modest PFS benefit with afatinib over methotrexate [54], this approach was abandoned owing to the lack of OS benefit and the development of more promising immunotherapies, to be discussed below.

### 4.2. Farnesyltransferase Inhibitors

Of the three RAS genes (*HRAS*, *KRAS*, and *NRAS*), *HRAS* is the most commonly mutated in HNSCC [12]. Thus, the RAS-RAF pathway is a target of high therapeutic interest. Since proper trafficking and localization of RAS proteins require several posttranslational modifications, a potential strategy to inhibit oncogenic RAS activity is to disrupt these posttranslational modifications such as RAS prenylation through inhibition of farnesyltransferase.

In February 2021, the FDA designated farnesyltransferase inhibitor (FTI) tipifarnib, still under investigation, as a breakthrough therapy for the treatment of *HRAS*-mutant R/M HNSC based on the outcomes of the RUN-HN trial, especially for patients whose disease had progressed while being treated with platinum-based chemotherapy [55]. Tipifarnib is a first-in-class nonpeptidomimetic quinolinone that binds to and potently inhibits FT. The phase-II RUN-HN (NCT02383927) trial included 30 patients with R/M *HRAS*-mutant HNSCC treated with tipifarnib and demonstrated a 50% ORR in the 18 evaluable patients. In addition, tipifarnib showed a median PFS of 5.9 months, as compared to 2.8 months with the last prior line of therapy, and the median OS was 15.4 months with tipifarnib [56]. Currently, tipifarnib is being studied in a pivotal phase-II AIM-HN study (NCT03719690), which will further assess the ORR of patients with *HRAS*-mutant HNSCC treated with tipifarnib [56]. Another trial is currently evaluating the combination of tipifarnib and the PI3K inhibitor alpelisib in R/M HNSCC with *HRAS* and *PIK3CA* alterations (NCT04997902) (clinicaltrials.gov).

### 4.3. PI3K/AKT/mTOR Inhibitors for PI3K-Mutant HNSCC

The PI3K/AKT/mTOR signaling pathway is crucial for various cellular processes, including cell growth and division, metabolism, and migration, all of which can be dysregulated in cancer. This pathway is one of the most frequently activated ones in HNSCC, with activation in more than 90% of HNSCCs [57,58], including both HPV-positive and HPV-negative subsets [12]. Oncogenic activation of the PI3K pathway can be caused by EGFR activation, PI3K overexpression, gain-of-function mutations and/or amplifications in *PIK3CA*, loss-of-function (LOF) mutations in *PTEN,* or oncogenic alterations in *AKT, TSC1/2*, *LKB1*, or *MTOR* [57,59]. Therefore, use of PI3K/AKT/mTOR inhibitors is an appealing therapeutic strategy for HNSCCs regardless of their HPV status. Of the several PI3K pathway inhibitors in different stages of clinical development, only a few have been approved by the FDA for the treatment of hematological cancers (copanlisib, idelalisib, umbralisib, and duvelisib), renal cell carcinoma (temsirolimus and everolimus) [59], and metastatic breast cancer (alpelisib) [60,61].

Numerous preclinical studies in HNSCC xenografts with *PIK3CA* mutations demonstrated the susceptibility of these tumors to PI3K/AKT/mTOR inhibitors [62,63,64,65,66,67,68,69], supporting their clinical development. However, clinical trials with pan-PI3K and dual PI3K/mTOR inhibitors have demonstrated only modest response rates that were not consistently better in *PIK3CA* mutant vs. wild-type (wt) tumors. One patient with endometrial cancer with *PIK3CA* and *PTEN* mutations had a complete response (CR) to the pan-PI3K inhibitor copanlisib in a phase-I study [70]. This outcome led to the initiation of the biomarker-based phase-II MATCH trial (NCT02465060), which examined copanlisib treatment in patients with advanced refractory solid tumors and with mutations in *PIK3CA* and *PTEN* and loss of *PTEN*. Buparlisib (BKM120), another pan-PI3K inhibitor, showed limited antitumor activity in patients with platinum-refractory R/M HNSCC, with a disease control rate of 49% and an ORR of only 3% (NCT01527877). There was no significant difference between cohorts with *PIK3CA*-mutant and non-mutant tumors in PFS (1.7 months vs. 1.8 months) or OS (3.4 months vs. 5.8 months (NCT01737450)) [71,72].

Additionally, use of the dual PI3K/mTOR inhibitor apitolisib [73] in patients with advanced solid tumors demonstrated modest clinical activity. Of the 14 evaluable patients with *PIK3CA* mutations, three patients showed partial responses (PRs), eight had stable disease, and three had progressive disease. Of the 120 enrolled participants in the study, 15 HNSCC patients were included independent of their *PIK3CA* status, of which only three had a PR. Similar response rates were observed with the dual PI3K/mTOR inhibitors gedatolisib [74] and bimiralisib (PQR309) [75] in patients with various advanced-stage cancers. Gedatolisib was administered to 77 patients with solid tumors, including four patients with *PIK3CA* alterations, who had stable disease for more than 6 months [74]. Likewise, a phase-I trial with bimiralisib evaluated 28 patients with advanced solid tumors, two of whom had *PIK3CA* mutations. One of them had stable disease and the other experienced a 26% reduction in tumor volume on treatment with bimiralisib [75]. 

Despite promising preclinical evidence, limited clinical activity and drug-related toxicities have hindered the use of pan-PI3K and dual PI3K/mTOR inhibitors for most solid tumors, leading to the development of isoform-selective PI3K inhibitors. Several ongoing clinical trials are investigating whether *PIK3CA*-mutant tumors are sensitive to drugs that inhibit the class-I PI3K catalytic subunit α isoform. Dose-escalation studies with the PI3Kα inhibitors taselisib, TAK-117, and alpelisib in patients with advanced-stage solid tumors have shown promising clinical activity in those harboring *PIK3CA* alterations as compared to patients with wt *PIK3CA* [76,77,78]. Taselisib treatment showed an ORR of 36% in patients with *PIK3CA* mutations but 0% in patients without *PIK3CA* mutations [76]. However, TAK-117 displayed a PR rate of 7.5% of patients with *PIK3CA*-altered tumors [77]. The phase-I trial with alpelisib specifically enrolled patients with *PIK3CA* mutations and showed an ORR of 6% and stable disease rate of 52%. Moreover, a dose-expansion arm of this study included four patients with *PIK3CA*-wt tumors, but they had no clinical benefit with alpelisib treatment [78]. Overall, PI3K isoform-specific inhibitors have demonstrated high rates of stable disease in *PIK3CA*-altered cancers and are better tolerated than pan-PI3K and dual PI3K/mTOR inhibitors. In addition, PI3Kβ-, PI3Kγ-, and PI3Kδ-specific inhibitors are in clinical development at various stages for use either as single agents or in combinations with chemotherapy, targeted therapy, or radiation therapy in R/M HNSCC (Table 1).

Although PI3Kα inhibitors demonstrated some clinical benefits in patients with *PIK3CA* mutations, they are insufficient to achieve a CR, and prolonged treatment results in therapy resistance. For this reason, development of this class of drugs as standalone modalities has not succeeded. In order to overcome therapeutic resistance and improve patient responses, PI3K pathway inhibitors are currently being evaluated for use in combination with other targeted therapies, chemotherapy, radiotherapy, and immunotherapies (Table 1). The addition of the pan-PI3K inhibitor PX-866 to docetaxel did not significantly improve the median PFS when compared to docetaxel alone in patients with R/M HNSCC (92 days; 95% CI, 46–119 vs. 82 days; 95% CI, 47–96; *p* = 0.42) or OS (263 days; 95% CI, 125–383 vs. 195 days; 95% CI, 121–NR; *p* = 0.62) [79]. Likewise, the combination of PX-866 with cetuximab had similar median PFS and OS durations when compared to cetuximab alone (PFS: 80 days in both groups; *p* = 0.48; OS: 211 days; 95% CI, 149–279 in the combination group vs. 256 days; 95% CI, 148–NR; *p* = 0.6) in the cetuximab-alone group. More importantly, patients harboring *PIK3CA* mutations did not have any response to combined PX-866 and cetuximab treatment [80]. The BERIL-1 phase-II trial in patients with platinum-refractory R/M HNSCC showed a modestly longer median PFS in patients receiving a combination of buparlisib and paclitaxel (4.6 months; 95% CI, 3.5–5.3) than in the paclitaxel and placebo group (3.5 months; 95% CI, 2.2–3.7; *p =* 0.011) (NCT01852292) [81]. A phase-III trial of this combination in R/M HNSCC is ongoing (NCT04338399). A phase-I trial with copanlisib and cetuximab (NCT02822482) in patients with R/M HNSCC in whom platinum or cetuximab therapy failed was stopped early due to unfavorable toxic effects in the combination arm [82]. Overall, these trials showed limited clinical efficacy and significant drug toxicity in the combination groups, emphasizing the need for better biomarkers of sensitivity.

However, in contrast to the results in R/M HNSCC, alpelisib in combination with drugs targeting the estrogen receptors has shown robust responses compared to single-agent drugs, leading to the FDA approval of alpelisib for the treatment of *PIK3CA*-mutant metastatic breast cancer [60,61]. Two phase-II clinical trials are currently evaluating the clinical efficacy of alpelisib as monotherapy in HPV-positive HNSCC (NCT03601507) and alpelisib in combination with the farnesyltransferase inhibitor tipifarnib in *HRAS-* and *PIK3CA*-mutant HNSCC (NCT04997902). 

*PIK3CA* gene mutations are classified as canonical or noncanonical. Canonical mutations are the most common, occurring in one of three hotspot locations (E542, E545, and H1047) of the p110α subunit and leading to the activation of the PI3K pathway. Noncanonical mutations are rare, distributed throughout the p110α subunit, and may be activating or non-activating. Moreover, around 27% of the noncanonical mutations characterized in TCGA are unique to HNSCC and have not been identified in other cancers with *PIK3CA* alterations [83]. Alpelisib, in addition to being effective against canonical *PIK3CA* alterations, has shown efficacy in preclinical models with three frequently occurring noncanonical mutations [61,84]. Moreover, a recent study also reported a remarkable response (73% tumor shrinkage) in an HNSCC patient with a noncanonical activating *PIK3CA* mutation who received alpelisib monotherapy [83], underscoring the need to further understand the noncanonical *PIK3CA* mutation biology. 

Despite ongoing clinical trials of PI3K inhibitors, their clinical translation has been challenging owing to their modest efficacy as single agents, their unfavorable toxicity profiles, the emergence of resistance, and the lack of a biomarker-selective approach for treatment. In retrospect, these disappointing clinical studies may have been predicted by preclinical studies that demonstrated that although *PIK3CA*-mutant HNSCC cell lines are more sensitive to PI3K inhibitors than *PIK3CA*-wt HNSCC cell lines [62,63], PI3K inhibition leads only to growth arrest, not cell death [85].

### 4.4. PI3K Inhibitors for NOTCH1-Mutant HNSCC

*NOTCH1* is one of the most frequently mutated genes in HNSCC [12]. Functional characterization has confirmed structural predictions that most of the *NOTCH1* mutations in HNSCC are LOF mutations, supporting a tumor-suppressor function for *NOTCH1* in HNSCC [86]. The occurrence of these mutations may predict a response to ICI and PI3K inhibitors [86]. Previous research from our group used an unbiased pharmacogenomics approach and identified a remarkable correlation between *NOTCH1* LOF mutations and sensitivity to PI3K inhibitors in HNSCC cell lines [87]. Testing of a panel of seven different PI3K pathway inhibitors in 59 HNSCC cell lines showed that the *NOTCH1* mutation correlated with inhibitor sensitivity. PI3K inhibition led to more apoptosis in *NOTCH1*-mutant than in *NOTCH1*-wt HNSCC cell lines in vitro. Similarly, *NOTCH1*-mutant HNSCC xenografts treated with PI3K pathway inhibitors demonstrated elevated cell death and significant tumor volume reduction in vivo [87].

This promising preclinical evidence was bolstered by findings from two independent studies reporting that vulnerability to PI3K inhibition was conferred by *NOTCH1* mutations. PX-866 led to significant tumor reduction in two *NOTCH1*-mutant HNSCC patient-derived xenograft models [63]. Activation of the NOTCH signaling pathway in breast cancers conferred resistance to PI3K/mTOR inhibitors [88]. One patient with heavily pretreated R/M HNSCC with a LOF *NOTCH1* mutation was enrolled in a phase-I study of bimiralisib and had a PR (85% reduction in target lesion) that was sustained for 36 weeks [89]. This observation was later tested in a phase-II trial (NCT03740100) [90] that showed modest single-agent activity of bimiralisib in *NOTCH1*-mutant HNSCC. Patients treated with bimiralisib had better outcomes (ORR: 17%, OS: 7 months, and PFS: 5 months) than historical control patients treated with standard therapies (ORR: 5.8%, OS: 5.1 months, and PFS: 2.7 months). Overall, these studies support *NOTCH1* LOF mutations as a predictive biomarker for the sensitivity to PI3K pathway inhibitors in HNSCC.

Mechanistic studies revealed a differential protein expression profile between *NOTCH1*-mutant and -wt HNSCC cells when treated with PI3K inhibitors. PDK1, a downstream signaling molecule in the PI3K pathway [87], and Aurora kinase B were exclusively and significantly downregulated in *NOTCH1*-mutant HNSCC on PI3K inhibition [91,92]. As a result, depleting cellular levels of *NOTCH1* [87], PDK1 [87], or Aurora kinase B [93] in *NOTCH1*-wt HNSCC cells sensitized them to PI3K/AKT/mTOR inhibition, leading to cell death. Since PI3K pathway inhibitors have modest single-agent activity, these mechanistic studies may identify effective combinatorial approaches that are superior to monotherapy and may also overcome the innate and acquired resistance that may develop with the use of targeted therapy.

### 4.5. Aurora Kinase Inhibitors

Aurora kinases are serine-threonine kinases that play an important role in cell-cycle regulation. These kinases help in the regulation of cell division, most importantly promoting the entry into mitosis, centrosome maturation, microtubule spindle assembly, and completion of cytokinesis. Overexpression of the Aurora kinases induces aneuploidy and genomic instability, which are frequently observed in many tumors [94]. McMillan’s group was the first to demonstrate that HPV-transformed cells were sensitive to the Aurora A kinase inhibitor alisertib [95]. Using HPV-transformed cells, they showed that alisertib led to mitotic delay, polyploidy, and apoptosis in vitro and decreased tumor size in vivo [96,97,98]. In addition to HPV status, our in vitro studies [99] have shown that protein levels of Rb predict the response of squamous cancer cells to Aurora kinase inhibitors. Manipulating Rb protein expression altered the sensitivity to Aurora kinase inhibitors in HNSCC and other cancer types [100,101,102]. In addition, we observed that inhibition of Rb upregulated mitotic checkpoint complex (MCC) genes and resulted in chromosomal instability and prolonged mitosis. Our recent study has demonstrated that the combination of depletion of the MCC protein TRIP13 with low alisertib concentrations selectively enhanced cell death in HPV-positive, but not HPV-negative, HNSCC cells.

Substantial preclinical data targeted toward Aurora kinases have resulted in the development of multiple small molecule inhibitors. Drugs that are being tested in clinical trials are the pan-Aurora kinase drugs AMG 900, ilorasertib (ABT348), and danusertib (PHA-739,358); the Aurora kinase A- and B-targeted drugs AT-9283 and TT00420; the Aurora kinase B-targeted drugs barasertib (AZD1152) and chiauranib; the Aurora kinase A-specific drugs alisertib (MLN8237) and ENMD2076. Alisertib is being widely evaluated in various tumor indications, with seven ongoing early-phase clinical trials including a phase-I/II trial in Rb-deficient HPV-positive HNSCC in combination with a PD-1-inhibitor (NCT04555837).

### 4.6. FGFR Inhibitors

The FGFR family consists of four transmembrane receptor tyrosine kinases, FGFR1–4, which are activated by 18 fibroblast growth factor (FGF) ligands [103]. FGFR1–3 are commonly amplified and overexpressed in HNSCC [104,105], with 10% to 17% of HNSCC tumors having recurrent *FGFR1* amplifications [106,107]. FGFR3-activating mutations are present in 11% of HPV-positive oropharyngeal SCCs (OPSCCs) [29,108]. FGFR fusions are also found to be strong predictors of a response; however, the frequency of FGFR3-TACC3 fusions is lower in HNSCC patients (0.7%) [12].

FGFR inhibition is effective for some cancers with FGFR alterations. The FGFR-TKI erdafitinib was approved for patients with advanced urothelial cancers harboring specific FGFR genomic alterations [109]. Dumbrava et al. [110] showed a CR to an FGFR inhibitor in a patient with R/M HNSCC with FGF amplifications. A small phase-I study (n = 10) showed a disease control rate of 80% with the FGFR TKI rogaratinib [111]. Several ongoing clinical studies with a variety of FGFR inhibitors in cancers with FGFR genomic alterations are open to HNSCC patients. The outcomes of these trials will provide insights into the clinical efficacy of FGFR inhibition in HNSCC.

### 4.7. Epigenetic Targeted Inhibitors

Certain epigenetic modifications—that is, changes in gene expression that do not involve DNA sequence changes—play a crucial role in cancer development and the interaction between tumor cells and their microenvironment [112]. At present, nine FDA-approved agents, representing four epigenetic targets (DNMT, HDAC, IDH, and EZH2), are used for the treatment of a variety of cancers, and several drugs that target epigenetic mechanisms are currently in clinical trials [113]. Many studies have reported that HPV-negative HNSCC has lower DNA methylation levels than HPV-positive HNSCC, which harbors distinctly hypermethylated genomes [114]. Thus, researchers expect that HPV-positive HNSCC will have a more robust response to epigenetic targeted therapy.

Rodriguez et al. [115] investigated the combination of the HDAC inhibitor vorinostat with the PD-1-inhibitor pembrolizumab in patients with PD-L1-positive, immunotherapy-naive R/M HNSCC. This trial found higher response rates (32%) in the combination-treated cohort than in the control (20%) group treated with single-agent anti-PD-1 mAbs. These results point to a need for further clinical investigation in a phase-II study. A patient cohort preselected for PD-L1 expression would also be needed in further assessments of the combination treatment.

Regarding DNMT inhibitors, a phase-I trial of the use of the DNMT inhibitor azacytidine as neoadjuvant monotherapy, involving a phase-IB trial of decitabine monotherapy, is currently underway in patients with HPV-positive HNSCC. In addition, the trial is also examining R/M HNSCC patients’ use of decitabine in combination with PD-L1 inhibitor durvalumab, regardless of HPV status. Results from this study in patients with ICI refractory HNSCC are pending (NCT03019003).

### 4.8. VEGF Inhibitors

Vascular endothelial growth factor (VEGF) inhibitors are used to block angiogenesis in several solid tumors, such as colorectal, renal cell, ovarian, gastric, and thyroid cancers [116]. FDA-approved bevacizumab is a VEGF-targeted mAb used to treat numerous cancers, either as a single agent or in combination with chemotherapy or radiotherapy [116]. In vitro data in HNSCC showed decreased VEGF secretion [117], reduced tumor growth, and enhanced cancer cell death when bevacizumab was used with radiotherapy [118]. In a phase-III trial in R/M HNSCC, adding bevacizumab to platinum-based chemotherapy significantly improved both the PFS (*p* = 0.0014) and ORR (*p* = 0.016), although no significant improvement in OS was found [119]. Moreover, the bevacizumab-chemotherapy combination was associated with significantly more treatment-related grade 3–5 bleeding events (6.7% vs. 0.5%; *p* < 0.001) and treatment-related deaths (9.3% vs. 3.2%; *p* = 0.022) than chemotherapy alone [120].

Studies of VEGFR inhibitors in R/M HNSCC showed modest effects. Vandetanib, a TKI that inhibits both EGFR and VEGFR, resulted in an ORR of 13% (PR in 2/15 patients) when combined with docetaxel in platinum-resistant R/M HNSCC [121]. The VEGFR TKIs sorafenib and sunitinib were both well-tolerated but also had only modest therapeutic effects [122,123]. Given their modest activity and considerable toxicity, most VEGFR inhibitor development has been halted for R/M HNSCC, with the exception of lenvatinib, which is being tested in combination with immunotherapy (described below).

### 4.9. IAP Inhibitors

Cellular inhibitor of apoptosis (cIAPs) and X-linked IAP (XIAP) proteins are both negative regulators of caspase-mediated apoptosis, and additionally, XIAPs also regulates mitochondria-mediated apoptosis [124]. A potent, orally active, small-molecule IAP inhibitor, Debio 1143 (AT-406, SM-406, xevinapant), may promote apoptosis in tumor cells by blocking both XIAP and cIAPs via restoration of caspase activity [125]. Sun and colleagues investigated the efficacy of Debio 1143 in a phase-II study evaluating 96 patients who received chemoradiotherapy (high-dose cisplatin and concurrent radiotherapy) with or without Debio 1143 (NCT02022098) [125]. The median PFS in the group that received chemoradiotherapy alone was 16.9 months; the median PFS was not reached in the group that also received Debio 1143 (*p* = 0.0069). At 2 years, the chemoradiotherapy + Debio 1143 group had a PFS rate of 72%, while the chemoradiotherapy-only group had a PFS rate of 41% (*p* = 0.0026). In addition, the chemoradiotherapy + Debio 1143 combination had a favorable safety profile at 2 to 3 years of follow-up [126]. This was the first treatment regimen to have better efficacy than chemoradiotherapy in a randomized trial. In February 2020, Debio 1143 was designated as a breakthrough therapy by the FDA for the treatment, in combination with chemoradiotherapy, of patients with locally advanced HNSCC that were previously untreated and unresectable. A phase-III trial of Debio 1143 is currently ongoing in combination with chemoradiotherapy and is expected to enroll about 700 patients (NCT04459715). These data suggest that inhibition of IAPs is yet another novel and promising approach for patients with high-risk locally advanced HNSCC. However, all relevant current trials are focused on locally advanced HNSCC, with none in the R/M HNSCC setting.

### 4.10. STAT3 Inhibitors

Several lines of evidence support a role for a signal transducer and activator of transcription 3 (STAT3) in HNSCC progression and survival [127]. Activated STAT3, defined as phosphorylated or nuclear STAT3, is found in 37% to 75% of HNSCC tumors and associated with a more advanced disease stage and poor survival [128,129,130]. Likewise, there are increased levels of phosphorylated STAT3 in most HNSCC cell lines [131]. HNSCC preclinical models depend on constitutively activated STAT3 for proliferation and survival [129,132,133]. The small-molecule STAT3 inhibitor TTI-101 inhibited anchorage-dependent and -independent growth of multiple human HNSCC cell lines in vitro and reduced tumor growth of radiation-resistant human HNSCC xenografts in vivo [131].

In addition to its direct effects on cancer cells, STAT3 also contributes to tumor vascularization and cancer immune evasion [134] by inhibiting the maturation of dendritic cells [135,136,137,138] and stimulating immunosuppressive cells in the tumor microenvironment, including myeloid-derived suppressor cells [139,140,141], M2 macrophages [135,142,143], T-helper 17 cells [144,145,146], and regulatory T cells [135]. The combination of the antisense STAT3 oligonucleotide danvatirsen with α-PD-L1 therapy (durvalumab) in HNSCC patients improved response rates over those in historic controls treated with α-PD-L1 therapy alone [147]. This result suggests that STAT3 inhibition may synergize with ICI in R/M HNSCC. Inhibiting STAT3 may also reduce immune-related adverse events [148]. 

However, inhibiting STAT3 in patients has been a challenge. The small-molecule STAT3 inhibitor TTI-101 is currently undergoing phase-I testing. No currently recruiting trials are investigating the most advanced of the STAT3 antisense oligonucleotides, danvatirsen (IONIS-STAT3-2.5Rx, AZD9150). The recent development of proteolysis-targeting chimera (PROTAC) drugs offers a promising future strategy for specifically inhibiting STAT3 in humans [127].

### 4.11. Antibody-Drug Conjugates

Antibody-drug conjugates (ADCs) are complex targeted agents that are composed of an antibody attached to a cytotoxic drug. This has shown promise in the treatment of certain cancers, including acute leukemia, breast cancer, cervical cancer, and Hodgkin’s lymphoma. Since they are delivered locally, ADCs are expected to be more effective and safer compared to standard chemotherapy [149]. ADCs are targeted to tissue factors expressed on tumor cell surfaces so that they can deliver a toxic payload to these cells [150]. Tisotumab vedotin is an ADC consisting of a human mAb that binds to tissue factor-011 and a microtubule-disrupting agent, monomethyl auristatin E. The FDA has granted accelerated approval to tisotumab vedotin for the treatment of adults with R/M cervical cancer that has progressed during or after chemotherapy. The drug was approved based on data points from the innovaTV 204 trial in a phase-II setting (NCT03438396), where 101 patients were treated and demonstrated an ORR of 24% (95% CI, 15.9–33.3%), with a CRR of 7% and median duration of response of 8.3 months (95% CI, 4.2 months–NR) [151]. Preliminary data from the phase-II innovaTV 207 trial (NCT03485209), which enrolled HNSCC patients who experienced disease progression after treatment with platinum chemotherapy and an ICI, demonstrated antitumor activity and a manageable safety profile. Patients receiving tisotumab vedotin alone had an ORR of 16%, median PFS of 4.2 months, and median OS of 9.4 months [152].

## 5. Immunotherapy for Head and Neck Cancer

Immunotherapy can be an effective treatment option for HNSCC given the nature of enhanced mutation in tumor cells, which leads to immune cell infiltration [153,154]. The responses of ICIs are dependent on tumor-derived T-cells [155]. The FDA approved the use of the PD-1 mAb nivolumab for the treatment of platinum-resistant R/M HNSCC in 2016 and the use of pembrolizumab as a frontline treatment for HNSCC in 2019, markedly altering the landscape of standard HNSCC therapy and engendering a major shift toward immunotherapy as a focus of future therapy development.

### 5.1. PD-1/PD-L1 Inhibitors

PD-1 is an immune-response suppressor primarily expressed on immune cells such as T and B lymphocytes, dendritic cells, and myeloid cells. Binding with the ligands, PD-L1 or PD-L2 activates PD-1, and prolonged PD-1 activation impairs and exhausts the immune response. HNSCC cells, among many other types of cancer cells, express PD-L1 [156,157]. PD-L1 expression inhibits the anticancer responses of tumor-infiltrating lymphocytes (TILs) and allows tumor cells to evade immune surveillance.

In the phase-IB KEYNOTE-012 study, patients with R/M HNSCC who received pembrolizumab had an ORR of 18% and 6-month PFS rate of 23% [158,159,160]. In the single-arm, phase-II KEYNOTE-055 study, patients with R/M HNSCC that was resistant to both platinum agents and cetuximab received pembrolizumab. The ORR was 16%, median PFS was 2.1 months, and median OS was 8 months [161].

The CheckMate 141 randomized phase-III trial compared nivolumab to the standard of care in patients with R/M HNSCC who had disease progression within 6 months of receiving platinum-based chemotherapy [162]. The ORR was 5.8% for standard care and 13.3% for nivolumab. The primary endpoint, OS, was significantly better in the nivolumab group than in the standard-care group (7.7 months vs. 5.1 months, *p* = 0.01) [163]. These results made nivolumab a standard treatment option for platinum-resistant R/M HNSCC. At the 2-year follow-up, CheckMate 141 showed a sustained OS advantage in patients treated with nivolumab compared to those treated with standard care (16.9% vs. 6%) [163].

KEYNOTE-048 was a randomized, phase-III trial for previously untreated R/M HNSCC (n = 882) with three arms: pembrolizumab alone, pembrolizumab plus chemotherapy (5-fluorouracil and a platinum-based agent), and chemotherapy plus cetuximab. Outcomes were reported for the total study population and for groups with PD-L1 combined positive scores (CPS) of 20 or higher and 1 or higher. Pembrolizumab alone was compared to chemotherapy plus cetuximab. Pembrolizumab plus chemotherapy was compared to chemotherapy plus cetuximab.

In the total study population, patients treated with pembrolizumab plus chemotherapy had a longer median OS than patients treated with the prior standard of care, chemotherapy plus cetuximab (13 vs. 10.7 months; *p* = 0.034). The median OS for the pembrolizumab-alone group was longer than that for the chemotherapy plus cetuximab group in both the CPS ≥ 20 (14.7 vs. 11 months; *p* = 0.004) and CPS ≥ 1 (13.6 vs. 10.4 months; *p* = 0.001) subgroups [164]. Based on KEYNOTE-048, pembrolizumab was approved by the FDA for use as the first-line therapy (in combination with chemotherapy) in all patients with R/M HNSCC. It was also approved as a single agent in those with CPS ≥ 1.

A recent post-hoc analysis compared the CPS <1 (n = 128) and CPS 1–19 (n = 373) subgroups [165]. In the CPS < 1 subgroup, the median OS duration of patients treated with pembrolizumab was shorter than in patients treated with chemotherapy plus cetuximab (7.9 vs. 11.3 months; HR, 1.51; 95% CI, 0.96 to 2.37). In the CPS 1–19 subgroup, however, the median OS was slightly longer in the pembrolizumab group (10.8 vs. 10.1 months; HR, 0.86; 95% CI, 0.66 to 1.12). The median OS duration for the group with CPS < 1 receiving pembrolizumab plus chemotherapy was 10.7 months, as compared to 11.3 months in the group receiving chemotherapy plus cetuximab (HR, 1.21; 95% CI, 0.76 to 1.94). In the group with CPS 1–19, the median OS was 12.7 months for the pembrolizumab plus chemotherapy group and 9.9 months for the chemotherapy plus cetuximab group (HR, 0.71; 95% CI, 0.54 to 0.94).

Despite the clear benefits of ICI, only a subset of patients with R/M HNSCC benefitted from it. Expression of PD-L1 is the best-studied predictive biomarker for PD-1/PD-L1 inhibitors. The number of PD-L1-positive cells (including tumor cells, lymphocytes, and macrophages) compared to the total number of viable tumor cells, referred to as the CPS value, when known in the HNSCC setting, allows for a better response to immunotherapy compared to analyzing PD-L1 expression alone [166]. Around 50% to 60% of HNSCC tumor cells express PD-L1 (i.e., the total positive score) [167], but when infiltrating immune cells are included in the measurement (i.e., CPS), the percentage of PD-L1-positive cells increases to 85%. Subgroup analysis of KEYNOTE-048 patients confirmed better responses to ICI with increasing CPS scores.

There are several other biomarkers that have been studied but not fully validated yet, which can contribute toward better response prediction to anti-PD1 therapy, for example, expression of PD-L2 [168]. KEYNOTE-012 data revealed that PD-L2 protein expression helped in predicting the response to anti-PD-1 therapy irrespective of expression of PD-L1 [169]. Another example of a biomarker helping in predicting the response to anti-PD1 therapy can be attributed to the HPV status, as well. In the KEYNOTE-012 trial, patients with PD-L1-positive/HPV-positive disease had a better ORR than PD-L1-positive/HPV-negative disease, with 25% and 14% ORR, respectively. In this case, from the overall patient pool, 62% patients had a HPV-negative status, whereas 38% were HPV-positive [158]. Subsequently, another trial (KEYNOTE-048) showed that HPV-positive patients displayed better results for combination therapy involving pembrolizumab plus chemotherapy, thereby enabling such patients to potentially receive combination therapy [170]. A number of trials are currently underway to evaluate several different combinations using ICI, therapeutic vaccines, co-stimulatory agonists, and targeted and cytotoxic agents (Table 2).

### 5.2. Vaccines

HPV-positive HNSCC contains E6 and E7 oncoproteins that are recognized as non-self-antigens required to maintain the malignant phenotype [171]. HPV-targeted therapeutic vaccines work against HPV infection by inducing a T-cell response. There are FDA-approved HPV vaccines available in the market to protect from high-risk HPV infection and cancer caused by them [172], but these vaccines do not treat established cancer.

E6/E7 antigen-targeting vaccines demonstrated efficacy in patients with HPV-induced cervical intraepithelial neoplasia and are now being evaluated in an R/M HNSCC setting [173]. However, vaccines targeting HPV16 have, by themselves, failed to treat recurrent advanced HPV-positive cancers. Therefore, combinations of ICI with therapeutic HPV vaccines are being tested for recurrent advanced HPV-positive cancers. A phase-IB/II trial (NCT03260023) of an HPV16 E6/E7 and IL-2 vaccine (TG4001) plus a PD-L1 inhibitor in patients with R/M HPV16-positive cancers that did not respond to available standard treatments demonstrated an ORR of 23.5% [174]. The vaccine-immunotherapy combination induced an HPV16 E6/E7-specific T-cell response and increased the number of TILs in the tumor microenvironment.

A combination of a synthetic long-peptide HPV16 vaccine (ISA101) with nivolumab, in a phase-II study, showed promising data with a 33% ORR and median OS duration of 17.5 months (NCT02426892) in patients with HPV16-positive R/M HNSCC [175]. In addition, two more phase-II trials of ISA101 are underway, one with utomilumab, a CD137 agonist, for patients with incurable HPV16-positive OPSCC (NCT03258008) [176], and the other with cemiplimab in R/M HPV16-positive OPSCC.

ADXS11-001 (axalimogene filolisbac) is a live-attenuated vaccine encoding an HPV16 E7 oncoprotein. It is currently in phase-II trials evaluating the HPV-specific T-cell response and safety in a neoadjuvant setting in HPV-positive OPSCC (NCT02002182). DPX-E7 is a synthetic peptide-based vaccine targeting HPV16 E711-19, which is under investigation in an open-label phase-IB/II trial. This trial, the results of which have not yet been released, includes HLA-A*02-01 patients with HPV16-associated OPSCC, anal cancer, and cervical cancer (NCT02865135).

Studies have shown that the E6 oncoprotein regulates telomerase reverse transcriptase (TERT) gene transcriptional activation, leading to its overexpression on cancer cell surfaces and thereby cell proliferation. A vaccine containing telomerase-derived cancer peptides, UCPVax, has been designed to activate CD4+ T-helper cells against telomerase-expressing cells. It is currently in a phase-II VolATIL trial (NCT03946358) in combination with atezolizumab in locally advanced or metastatic HPV-positive patients.

Nucleic acid-derived vaccines, such as DNA and RNA vaccines, are easier to synthesize and develop compared to peptide vaccines. MEDI0457, a DNA vaccine, was evaluated in a phase-IB/II trial enrolling 22 patients with locally advanced HPV16/HPV18-positive HNSCC. The tumor regression rate was 50%, and an increase in T-effector cells could overcome HPV-driven tumor immune evasion. Currently, MEDI0457 is being evaluated in combination with the PD-L1 inhibitor durvalumab in a phase-IB/IIA study (NCT03162224) of 35 patients with HPV-associated R/M HNSCC. BNT113 is an RNA lipoplex-based mRNA vaccine encoding HPV16 E6 and E7 that is being evaluated in combination with pembrolizumab in HPV-positive and PD-L1-expressing HNSCC (NCT04534205). In the future, HPV-directed therapeutic vaccines may move beyond targeting E6 and E7 to include the viral E1, E2, E4, E5, and L1 proteins [177].

In addition to HPV vaccines, recent advances in NGS technologies have led to better identification of neoantigens, thereby boosting the development of cancer vaccine research and strategies. Cancer vaccination enhances tumor-specific CD4+ and CD8+ T cells, which eliminates tumor cells without affecting the normal cells. Currently, two neoantigen vaccine trials are ongoing. First, there is a combination of personalized cancer vaccination with an anti-PD-1 mAb in a phase-IB setting, focused on advanced HNSCC patients. The tumor-derived neoantigens would be specific to an individual and tumor (NCT03568058) in this study. Second, there is a phase-I trial with HNSCC patients with progressive disease after anti-PD-1 or anti-PD-L1 treatment. The patients in the trial have been termed as requiring a “personalized and adjusted neoantigen peptide vaccine” (PANDA-VAC); they will be vaccinated with six neoantigens alongside treatment with pembrolizumab (NCT04266730). 

### 5.3. Adoptive Cellular Therapy

Chimeric antigen receptor (CAR)-T cell therapy is a popular treatment option for hematologic malignancies [178]. CAR-T cells use activated cytotoxic T lymphocytes that are primed to patient-specific antigens to cause tumor cell death. There have been several CAR-T-based cell therapy trials against different antigens for HNSCC. For example, T4 immunotherapy, a CAR-T cell therapy primed to antigens for the ErbB family (EGFR, HER2-4), which is highly upregulated in HNSCC [179], was studied in a dose-escalation phase-I study for locally advanced or recurrent HNSCC. The CAR-T cell doses were increased from 1 × 10^7^ to 10 × 10^7^ T4+ T cells in different patient cohorts. After 6 weeks, patients who received the highest dose of 10 × 10^7^ T4+ T cells displayed stable disease. The overall disease control rate was 69%, despite the patients having had rapidly progressing tumors on trial entry [179]. 

In addition to T4 CAR-T cell therapy, recent research has focused on developing effective HPV-targeted T-cell receptors (TCRs). This therapeutic approach involves introducing a tumor antigen to alter or modify the genetics of T-cells. The genetically altered T-cells express receptors targeting a restricted epitope (HLA-A*02:01) of E6 TCR T cells. This has been investigated for treatment of patients with metastatic HPV16-positive HNSCC [180]. Initial results from a phase-I/II trial carried out in a pool of 12 patients with various metastatic, HPV16-positive, HLA-A*02:01-positive cancers showed that two patients had partial responses to this therapy. E6 TCR memory T cells were observed in these two patients at a high percentage after one month of treatment and ~7% after 10 months of treatment. However, no levels of E6 TCR memory T cells could be observed in patients who did not respond to the therapy [181,182]. Other ongoing TCR therapy trials include: E7 TCR treatment in HLA-A*02:01-positive patients with relapsed/refractory HPV16-positive cancers (KITE-439, NCT03912831); HPV-E6-specific TCR T-cell transfer in HPV-positive HNSCC (NCT03578406); autologous TIL infusion (LN-145/LN-145-S1) followed by IL-2 administration in patients with R/M HNSCC (NCT03083873).

### 5.4. Novel Immunotherapies

Apart from PD-1, other T-cell exhaustion markers are also targets for ICI and are upregulated in HPV-positive HNSCC, such as cytotoxic T-lymphocyte protein 4 (CTLA4), T-cell immunoreceptor with Ig and ITIM domains (TIGIT), and lymphocyte activation gene 3 protein (LAG3) [183,184].

Potential ICI resistance mechanisms in R/M HNSCC are diverse, and many molecular targeted agents are being tested in combination with ICIs. Potential combinations of ICI with Aurora kinase and STAT3 inhibitors are described above. Additional ICI-containing combinations are also being investigated based on the success of this strategy in other solid tumors. HPV-specific immunotherapies are further under development. The summary below is not exhaustive but focuses on the most promising or clinically advanced combinations.

Recently, a multi-kinase inhibitor against VEGFR1-3, lenvatinib, was tested with pembrolizumab and showed a 46% ORR in patients with advanced solid tumors [185]. Chen et al. [186] reported an ORR of 28.6% with a median OS of 6.2 months in a small cohort (n = 14) of patients with immunotherapy-refractory HNSCC. Two ongoing phase-II trials are testing combinations of lenvatinib with pembrolizumab for heavily pretreated R/M HNSCC patients. In addition, a study of a combination of pembrolizumab with the VEGFR2 inhibitor cabozantinib for patients with PD-L1-positive, CPS >1 R/M HNSCC resulted in an impressive ORR of 45% and an overall clinical benefit rate of 90%. The 1-year OS and PFS rates were 68% and 52%, respectively [187].

Previous studies have shown that PI3K pathway inhibition modulates the tumor microenvironment and has a direct effect on immune cells that could be used to improve patient responses [188]. Tumor hypoxia has been reported to be a resistance mechanism in R/M HNSCC patients treated with a PD-1 blockade. Preclinical studies in PD-1-inhibitor-resistant mouse models showed resistant tumors had high oxidative metabolism that led to increased intratumoral hypoxia and decreased CD8+ T cells [189]. Moreover, PI3K inhibitors induced a decrease in tumor hypoxia specifically in head and neck tumor xenograft models [190]. Therefore, the addition of PI3K inhibitors to ICI therapy could improve responses in patients who otherwise would not benefit from ICI therapy alone. However, combining PI3K inhibitors with ICI has been challenging as PI3K inhibitors may alter antitumor immunity or lead to immune-related adverse events. This obstacle could be overcome by adopting a modified treatment regimen with intermittent, rather than daily, dosing of PI3K inhibitors [191,192]. Notably, a phase-I/II trial with copanlisib combined with nivolumab and/or ipilimumab is ongoing in patients with advanced solid cancers who have genomic alterations in *PIK3CA* and *PTEN* (NCT04317105).

Two phase-II trials investigated the combination of cetuximab with either pembrolizumab or nivolumab [193,194]. The combination containing pembrolizumab showed promising results, with an ORR of 45% and median OS of 18 months in patients with platinum-refractory tumors who received no prior anti-EGFR therapy or immunotherapy. The nivolumab + cetuximab combination showed an ORR of 22% in a population with a high proportion of patients who had previously received cetuximab or ICI treatment [194]. Based on these data and the clear unmet need for therapy options in ICI-resistant R/M HNSCC, the combination of cetuximab and pembrolizumab is now included in the National Comprehensive Cancer Network guidelines [195].

EAGLE was a randomized phase-III study that compared the anti-PD-L1 mAb durvalumab plus the CTLA4 inhibitor tremelimumab to chemotherapy in R/M HNSCC. Durvalumab did not improve survival, either as a single agent or in combination with tremelimumab. Moreover, the HPV biomarker status did not predict a response to the combination [196]. CheckMate 651 was a phase-III trial comparing nivolumab plus ipilimumab with chemotherapy plus cetuximab as a frontline therapy in platinum-eligible patients with R/M HNSCC. The study did not meet the primary endpoint, but there were some positive observations in terms of OS for a small set of patients who had tumors that expressed PD-L1 with a CPS ≥ 20 [197]. Thus, current data do not support a role for CTLA4 inhibition in HNSCC.

A couple of other mAb combination-based treatments with PD-1/PD-L1 inhibitors are also currently under investigation, e.g., tiragolumab, humanized anti-TIGIT mAb, and relatimab, an anti-LAG3 mAb. Combination therapy involving tiragolumab and atezolizumab (PD-L1 inhibitor) is currently being evaluated as a frontline treatment for PD-L1-positive R/M HNSCC in the SKYSCRAPER-09 trial (NCT04665843). Along similar lines, a combination of relatlimab and nivolumab is also under trial for R/M HNSCC treatment (NCT04326257).

HPV16 oncoproteins in HPV-positive HNSCC have led to development of HPV-specific immunotherapies. One such example is a fusion protein, CUE-101, which selectively binds and activates HPV16 E7-specific CD8+ T cells, as shown in preclinical studies [198]. It is currently being evaluated in a dose-escalation and expansion phase-I study with or without a PD-1-inhibitor as a frontline treatment for patients with HPV16-positive R/M HNSCC (NCT03978689). Initial data showed increased tumor-infiltrating T cells (CD3+ GZMB+) in tumor tissue post-CUE-101 administration in one patient, and out of 33 patients, eight demonstrated no disease progression for 12 weeks or more [199].

In addition to activation of antigen-specific T cells, a newly developed treatment modality called near-infrared photoimmunotherapy (NIR-PIT) is currently being studied. It uses a mAb that is attached to IRDye700DX (IR700), a photo-absorbing dye, and activated by near-infrared light [200]. This causes the cells to die by swift swelling, blebbing, and rupturing, allowing the internal cell contents to come into contact with the extracellular compartment, leading to a strong immune response. A 50% ORR in previously inoperable HNSCC patients was observed in a phase-I/II trial of cetuximab with NIR-PIT (NCT02422979) [201]. NIR-PIT was approved as a treatment regimen for recurrent HNSCC in Japan given its high efficacy and low adverse event rate. A phase-III trial involving NIR-PIT is currently ongoing (active since 2018) for HNSCC patients with recurrent characteristics who have experienced treatment failure or tumor progression during or after at least two lines of therapy.

Additional approaches that are under investigation but beyond the scope of this review include oncolytic viruses, inhibitors of B7-H3 (enoblituzumab), inhibitors of IDO1 (epacadostat), NKG2A inhibitors (monalizumab), and definitive therapy for the treatment of oligometastatic disease.

## 6. Conclusions

The management of R/M HNSCC is rapidly evolving. Although current approaches to systemic therapies are limited to chemotherapy, anti-PD-1 immunotherapy, and cetuximab, multiple new approaches are currently under development and investigation. Progress in understanding the genomic landscape and tumor microenvironment of HNSCC has helped in delivering personalized and effective treatment approaches. For example, *HRAS*-mutant HNSCC responds to the farnesyltransferase inhibitor tipifarnib. Other promising targeted therapies include PI3K inhibitors for *NOTCH1*-mutant tumors and Aurora kinase inhibitors for Rb-deficient, HPV-positive HNSCCs. With recent approval of the ADC tisotumab vedotin for cervical cancer and promising phase-II results in HNSCC, it is highly possible that tisotumab vedotin will represent another option in R/M HNSCC. Recent advances in ICI have improved outcomes in both HPV-positive and -negative HNSCC. Advancements in data and research around molecular structures and immunological features have equipped us to differentiate between HPV-positive and -negative HNSCC and enabled us to create targeted therapeutic approaches as well as personalized medicine beyond ICI. Amongst these new opportunities, the most promising are NIR-PIT for localized recurrence; novel fusion proteins; anti-HPV therapeutic vaccines; HPV-specific adoptive T-cell therapies for HPV-positive R/M HNSCC; the combination of targeted therapies with ICIs. Future therapies for R/M HNSCC are likely to be directed toward specific patient subpopulations based on a better understanding of cancer biology.

## Figures and Tables

**Figure 1 ijms-23-07889-f001:**
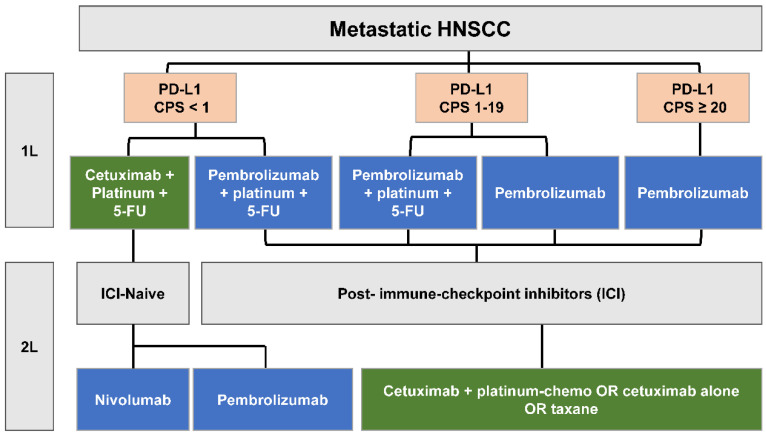
Standard-of-care treatment algorithm for metastatic HNSCC. 1L = first line, 2L = second line, CPS = combined positive score, 5-FU = 5-fluorouracil, ICI = immune-checkpoint inhibitor.

**Table 1 ijms-23-07889-t001:** Ongoing clinical trials with PI3K/AKT/mTOR inhibitors in HNSCC.

Class	Drug	Patient Cohort	Biomarker	Phase	Clinical Trials	Intervention	Status
Pan-PI3K inhibitor	Buparlisib (BKM-120)	R/M HNSCC	None	III	NCT04338399	Buparlisib + paclitaxel	Recruiting
		LA-HNSCC	HPV-positive	I	NCT02113878	Buparlisib with cisplatin + IMRT	Completed, Awaiting results
	Copanlisib (BAY 80-6946)	R/M HNSCC	None	I	NCT03735628	Copanlisib + nivolumab	Active, not recruiting
		R/M HNSCC	PIK3CA mutation, PTEN mutation/loss	II	NCT02465060	Copanlisib	Recruiting
Isoform-specific PI3K inhibitor	Alpelisib (BYL-719) (PI3Kα)	LA-HNSCC	HPV-positive	II	NCT03601507	Alpelisib	Recruiting
		R/M HNSCC	HRAS overexpression, PIK3CA mutation and/or amplification	I/II	NCT04997902	Tipifarnib + alpelisib	Recruiting
		R/M HNSCC	PI3K pathway alterations	II	NCT03292250	Alpelisib	Completed, awaiting results
		R/M HNSCC	None	II	NCT02145312	Alpelisib	Unknown
		R/M HNSCC	None	I	NCT01822613	Alpelisib + LJM716	Completed, awaiting results
		LA-HNSCC	None	I	NCT02282371	Alpelisib with cetuximab + IMRT	Completed, awaiting results
		LA-HNSCC	None	I	NCT02537223	Alpelisib with cisplatin + IMRT	Completed, awaiting results
	Duvelisib (VS-0145) (PI3K δ/γ)	R/M HNSCC	None	II	NCT05057247	Duvelisib + docetaxel	Recruiting
	GSK2636771 (PI3K β)	R/M HNSCC	PTEN mutation/loss	II	NCT02465060	GSK2636771	Recruiting
	Parsaclisib (INCB050465) (PI3K β)	R/M HNSCC	None	I	NCT02646748	Parsaclisib + pembrolizumab	Completed, awaiting results
	Serabelisib (INK-117) (PI3K α)	LA-HNSCC	PIK3CA mutation, KRAS mutation	I/II	NCT04073680	Serabelisib + canagliflozin	Unknown
	Taselisib (GDC-0032) (PI3K α/δ/γ)	R/M HNSCC	PIK3CA mutation, PTEN mutation/loss	II	NCT02465060	Taselisib	Recruiting
Dual PI3K/mTOR inhibitor	Gedatolisib (PF-05212384)	R/M HNSCC	PI3K pathway alterations	I	NCT03065062	Gedatolisib + palbociclib	Recruiting
AKT inhibitor	Ipatasertib (GDC-0068)	R/M HNSCC	AKT mutation	II	NCT02465060	Ipatasertib	Recruiting
		LA-HNSCC	None	I	NCT05172245	Ipatasertib with cisplatin + RT	Recruiting
		R/M HNSCC	None	II	NCT05172258	Ipatasertib + pembrolizumab	Recruiting
	Capivasertib (AZD5363)	R/M HNSCC	AKT mutation	II	NCT02465060	Capivasertib	Recruiting

AKT = AKT kinase; HPV = human papillomavirus; IMRT = intensity-modulated radiation therapy; mTOR = mammalian target of rapamycin; PI3K = phosphoinositide 3-kinase; LA-HNSCC = locally advanced head and neck squamous cell carcinoma; R/M HNSCC = recurrent and metastatic head and neck squamous cell carcinoma.

**Table 2 ijms-23-07889-t002:** Ongoing clinical trials evaluating novel immunotherapies in patients with recurrent/metastatic HNSCC.

Novel Immunotherapies in Combination with PD-1/PD-L1 Inhibitors, and Other Novel Checkpoint Inhibitor/Immunotherapies					
Drug(s)	Study Phase	Clinical Trials	Study Name	Intervention	HPV Status
Lenvatinib	III	NCT04199104	LEAP-10	Pembrolizumab vs. pembrolizumab + lenvatinib	HPV-positive
Bempegaldesleukin	II/III	NCT04969861	PROPEL-36	Bempegaldesleukin + pembrolizumab	
Nivolumab + ipilimumab	III	NCT03700905	IMSTAR-HN	Nivolumab + ipilimumab vs. surgery + RT	HPV-negative
Nivolumab	III	NCT03576417	NIVOSTOP	Nivolumab + RT + cisplatin vs. RT + cisplatin	unknown
Nivolumab + ipilimumab	III	NCT02741570	CheckMate 651	Nivolumab + ipilimumab vs. SOC (EXTREME regimen)	HPV-positive
Abemaciclib	I/II	NCT03655444		Abemaciclib + nivolumab	
Ramucirumab	I/II	NCT03650764		Ramucirumab + pembrolizumab	unknown
Duvelisib	I/II	NCT04193293		Duvelisib + pembrolizumab	unknown
Intratumoral MK-1454	II	NCT04220866		Intratumoral MK-1454 + pembrolizumab vs. pembrolizumab	unknown
Eftilagimod alpha	II	NCT04811027	TACTI-003	Eftilagimod alpha + pembrolizumab vs. pembrolizumab	HPV-positive
BNT113	II	NCT04534205	AHEAD-MERIT	BNT113 + pembrolizumab vs. pembrolizumab	HPV-positive
PDS0101 (HPV E6/E7 vaccine)	II	NCT04260126	VERSATILE002	Pembrolizumab + PDS0101 (HPV E6/E7 vaccine)	HPV-positive
Pepinemab	I/II	NCT04815720	KEYNOTE B84	Pepinemab + pembrolizumab	
Atezolizumab	II	NCT03818061	ATHENA	Atezolizumab + bevacizumab	HPV-positive
Avelumab	I	NCT03498378		Avelumab + Palbociclib + cetuximab	unknown
Alisertib	I	NCT04555837		Alisertib + pembrolizumab	HPV-positive
Cemiplimab	II	NCT04831450		Maintenance cemiplimab (anti-PD1)	
**Other Novel Checkpoint Inhibitors/Immunotherapy**					
Tiragolumab	II	NCT04665843	SKYSCRAPER-09	Tiragolumab + atezolizumab vs. atezolizumab	HPV-positive
Relatlimab	II	NCT04326257		Nivolumab + relatlimab vs. nivolumab + ipilimumab	unknown
Monalizumab	III	NCT04590963	INTERLINK-1	Monalizumab + cetuximab vs. cetuximab	
Epacadostat	I/II	NCT02327078	ECHO-204	Epacadostat + nivolumab	unknown
Enoblituzumab	I	NCT02475213	MGA271	Enoblituzumab + pembrolizumab	unknown
IMA201	I	NCT03247309		IMA201 (TCR-engineered in solid tumors, ACTengine)	
KITE-439	I	NCT03912831		KITE-439 (E7 T-cell receptor + cyclophosphamide + fludarabine)	HPV-positive
Autologous TILs	II	NCT03083873		Autologous TILs	HPV-positive

HPV, human papillomavirus; RT, radiotherapy; SOC, standard of care; TCR, T-cell receptor; TIL, tumor-infiltrating lymphocyte.

## Data Availability

Not applicable.

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
