# Peer review of "Novel Systemic Treatment Modalities Including Immunotherapy and Molecular Targeted Therapy for Recurrent and Metastatic Head and Neck Squamous Cell Carcinoma"

_ijms, 2022, doi:10.3390/ijms23147889_

Round 1

Reviewer 1 Report

The authors give a very broad review of the therapeutic landscape of R/M HNSCC. Beyond FDA-approved agents the ongoing investigations and pipelines are discussed. Some details are broader as the aim: non-R/M HNSCC or preclinical investigations are reported. For a better understanding, the relevance on R/M HNSCC could be more emphasized (e.g. as by IAP inhibitors). It would maybe more intuitive for the readers if the R/M HNSCC relevant information was edited to a new paragraph.

Line 25: please consider using a standard nomenclature - e.g. "PD-1-inhibitors"

Line 36: The discussed trials were focused on tumors with oral cavity, laryngeal and pharyngeal (some of them nasopharyngeal) origin. Are sinuses here really relevant to mention?

Lines 46-50: being a very rare genetic anomaly, FA causes only a very small proportion of HNSCC - if still deemed relevant for this introduction, then please consider refining, because not only the risk of oral cavitiy SCC, but all HNSCC is elevated.

Line 60: a phase III trial proved the benefit of oral screening among a high-risk group of tobacco, betel quid and/or alcohol users (Sankaranarayanan et al. 2005) - please consider revising your statement about effectiveness of screening

Line 139: You mean, p16 is used as a surrogate marker of HPV?

Lines 146-156: if you discuss the Bonner trial on LA-HNSCC, maybe you could also consider discussing and citing the EXTREME trial, published by Vermorken et al., which was the gold standard for R/M HNSCC for a decade. Separating LA-HNSCC & R/M HNSCC in two paragraphs could lead to better understanding.

Line 257: also radiation therapy

Line 310 "Table 1" - please consider specifying which of the trials are on R/M HNSCC vs. LA-HNSCC

Line 561: reference of CPS were "to all tumor cells" or "total tumor cell count"

Line 564: please consider using "infiltrating" instead of "surrounding"

Line 625: ... and combination of targeted therapies with ICIs

Author Response

July 8, 2022

Dear Editor and Reviewers,

Thank you for reviewing our manuscript about novel systemic treatment modalities for RM HNSCC.

We have addressed all the comments and provided our responses to each comment. We have made all our changes in the track change mode in the revised manuscript. We hope you find the revised manuscript appropriate for publication.

Sincerely,

Faye Johnson

Reviewer 2 Report

This manuscript that submitted by Ghosh et al. gathered recent molecular therapy and immunotherapy on head and neck squamous cell carcinoma (HNSCC) and summarize the status of these systemic treatment in clinical trial. Overall, this manuscript is well written in description of most top genes mutation, the discrepancy between HPV-positive and negative HNSCC, aberrant signaling pathway be used as target therapy by TKIs or immunotherapy, etc. The table 1 and 2 are useful for audiences to understand the status of PI3K-AKT inhibitors and immunotherapy in clinical development for treating HNSCC. This manuscript seems to reach the publishing criterion.

Author Response

As attachment.
